# ZIF-8@ZIF-67-Derived Co Embedded into Nitrogen-Doped Carbon Nanotube Hollow Porous Carbon Supported Pt as an Efficient Electrocatalyst for Methanol Oxidation

**DOI:** 10.3390/nano11102491

**Published:** 2021-09-24

**Authors:** Ruiying Wang, Mengran Lou, Jie Zhang, Zhipeng Sun, Zhiqian Li, Pengtao Wen

**Affiliations:** 1State Key Laboratory of Chemistry and Utilization of Carbon-based Energy Resource, Key Laboratory of Energy Materials Chemistry, Ministry of Education, Key Laboratory of Advanced Functional Materials, Autonomous Region, Institute of Applied Chemistry, College of Chemistry, Xinjiang University, Urumqi 830046, China; Loumr0315@126.com (M.L.); zj1994917@126.com (J.Z.); lzqdyx429146@163.com (Z.L.); w1437051215@126.com (P.W.); 2Materials and Energy School, Guangdong University of Technology, Guangzhou 510006, China

**Keywords:** metal-organic frameworks, N-doped carbon nanotube, hollow porous carbon, Pt-based catalysts, direct methanol fuel cells

## Abstract

It is of prime importance to develop anode electrocatalysts for direct methanol fuel cells (DMFCs) with good performance, which is critical for their commercial applications. Metal-organic framework (MOF)-derived carbon materials are extensively developed as supports of catalysts. Herein, Co embedded nitrogen-doped carbon nanotube hollow porous carbon (Co-NCNT-HPC) derived from MOFs have been fabricated, which were synthesized by pyrolyzing at an optimized temperature of 800 °C using ZIF-8@ZIF-67 as a precursor. The presence of ZIF-8@ZIF-67 ensures the doping of nitrogen and the large specific surface area of the support materials at high temperatures. A Pt/Co-NCNT-HPC800 sample, which was synthesized using Co-NCNT-HPC800 as a support, showed an enhanced mass activity of 416.2 mA mg^−1^_Pt_ for methanol oxidation reaction (MOR), and the onset potential of CO_ad_ oxidation of 0.51 V, which shifted negatively about 0.13 V compared with Pt/C (20%). Moreover, the Pt/Co-NCNT-HPC800 sample exhibits high stability. This work provides a facile strategy for MOF-derived carbon materials to construct advanced electrocatalysts for MOR.

## 1. Introduction

Electrochemical energy conversion storage based on fuel cells offers a viable solution in an era of world energy consumption and environmental degradation [1]. Direct methanol fuel cells (DMFCs) play an important role in energy conversion and storage [2,3]. Developing highly-active and low-cost anode catalysts for DMFCs is the main prerequisite for the commercialization of energy conversion systems [4,5]. Even though Pt is the most widely applied anode catalysts for DMFCs at present [6,7], its drawbacks of low electrocatalytic stability, high cost, and the easily poisoned by oxidation intermediates such as CO, have impeded its practical applications [8]. Many efforts have been made to commercialize Pt-based catalysts [9,10]; nevertheless, these anode catalysts for DMFCs still can’t meet the need for practical application. Therefore, it is very crucial to further improve the efficiency and performance of the Pt-based catalysts.

In our previous reports, this support played a significant part in the performance of catalysts, which is essential to enhance the mass and charge transfer of catalysts and improve the dispersion of metal particles [11,12,13]. The carbon-based catalysts with metal-organic frameworks (MOFs) consisting of metal nodes and organic linkers are a good strategy among many improved methods [14]. On the one hand, with the composition and structure diversities of MOFs, it is possible to prepare catalyst supports with ideal compositions and structures [15,16]. On the other hand, since the metal elements and organic compounds in MOFs can be converted in situ by pyrolysis to porous carbon supports with metal/metal oxide/metal-carbon, or even to porous carbon without metal or nitrogen doping, MOFs can also be selected as self-sacrificing precursors/templates for the preparation of nano porous carbon [17,18]. The MOF-derived porous carbon can inherit the structural properties and advantages of the corresponding MOFs [19]. Especially zeolitic imidazolium frameworks (ZIFs) are favorable precursor materials for nitrogen doping in the carbon base of MOF materials, as they contain enough nitrogen atoms in the high porous structures [20]. In previous studies, ZIF materials have been used to prepare carbon-based electrocatalysts [21,22,23]. The material of ZIF-67 has a zeolite-like three-dimensional topology, in addition to its intrinsic properties, it should be noted that 2-methylimidazole in ZIFs generates nitrogen-doped (N-doped) carbon during pyrolysis, which leads to enhanced mechanical strength, electron transfer rate, and its interaction with the electrolyte and reactants [24]. Consequently, MOF-derived N-doped carbon structures are expected to be promising precursors for the design of various porous carbon hybrid materials.

Although MOF-derived carbon materials can be N-doped in situ, thus improving the performance of the supports, increasing the electrochemical activities of the catalysts for methanol oxidation reaction (MOR) requires a further shortened ion transport path for the carbon supports to enhance the conductivity. To solve this problem, it is a good strategy to adjust the pore structures and enlarge the specific surface areas of the carbon supports. High-temperature pyrolysis can solve the above problems while preparing MOF-derived N-doped carbon materials [25,26]. Based on shortening the ion transport channels, the large specific surface areas expose more active sites for uniform Pt particle dispersions; therefore, adjusting the pore structures can improve the wettabilities of the carbon supports, and the electrolyte diffuses effectively during the electrocatalytic process [27,28,29]. The specific surface areas of MOF-derived N-doped carbon structures can be improved by ZIF-8 with the same series of porous MOF materials as ZIF-67 under the same conditions [30]. Therefore, the N-doped porous carbon materials with MOFs as precursors for electrochemical applications have great potential for development.

Herein, we developed Co embedded into nitrogen-doped carbon nanotube hollow porous carbon (Co-NCNT-HPC) materials derived from the pyrolysis of ZIF-8@ZIF-67, with the pyrolysis temperature optimized, which supports Pt-based composite catalysts that were synthesized. The as-prepared Pt-based catalyst of Pt/Co-NCNT-HPC800 showed enhanced mass activity, excellent CO poisoning resistance and outstanding stability for methanol electrooxidation owing to its compositional and structural features.

## 2. Materials and Methods

### 2.1. Chemicals

Chloroplatinic acid hexahydrate (H_2_PtCl_6_·6H_2_O,), 2-methylimidazole (MeIM) and cobalt nitrate hexahydrate (Co(NO_3_)_2_·6H_2_O) were purchased from Aladdin. Commercial Pt/C (20%, Johnson Matthey, London, UK), Nafion solution (5 wt%, DuPont, Wilmington, DE, USA), Methanol (Tianjin Fengchuan Fine Chemical Research Institute, Tianjin, China), Glycol (EG, Tianjin Guangfu Fine Chemical Co., Ltd., Tianjin, China), Ethanol (Tianjin Zhiyuan Fine Chemical Research Institute, Tianjin, China) and KOH (Tianjin Beilian Fine Chemical Research Institute, Tianjin, China) were analytically pure, and used as received without further purification. Deionized water was used for all experiments.

### 2.2. Synthesis of ZIF-8@ZIF-67

Zn(NO_3_)_2_·6H_2_O (5.95 g) and MeIM (6.16 g) was dissolved in 150 mL of methanol, respectively. Subsequently, two solutions were mixed, which were stirred for 24 h to produce a white solid product, then was washed by methanol and dried under vacuum for 8 h at 60 °C to obtain ZIF-8.

ZIF-8 (0.50 g), Co(NO_3_)_2_·6H_2_O (5.82 g), and MeIM (6.16 g) was dissolved in 100 mL of methanol, respectively. Then, Co(NO_3_)_2_·6H_2_O solution was poured into the ZIF-8 solution, followed by the addition of the MeIM solution to the above mixture, and was stirred for 24 h. The solution was washed in methanol and then dried under vacuum for 8 h at 60 °C to obtain ZIF-8@ZIF-67. The synthesized ZIF-8@ZIF-67 was held under Ar atmosphere for 4 h with a heating rate of 2 °C min^−1^ at different temperatures (600, 700, 800, and 900 °C). A series of Co-NCNT-HPC materials were obtained that were named Co-NCNT-HPC600, Co-NCNT-HPC700, Co-NCNT-HPC800, and Co-NCNT-HPC900. Comparison of these materials considers the whole process without adding ZIF-8, only the pure ZIF-67 is pyrolyze under Ar atmosphere at 800 °C for 4 h, with this porous carbon material obtained called Co-PC800 [13].

### 2.3. Synthesis of Pt/Co-NCNT-HPC

Pt-based composite catalyst of Pt/Co-NCNT-HPC was synthesized by a thermal reflux method. Co-NCNT-HPC (0.04 g) support was added to 25 mL EG, ultrasound for 30 min, and then 1 mL of 0.05 mol L^−1^ H_2_PtCl_6_-EG solution and 0.4 mL of 0.4 mol L^−1^ KOH aqueous solution were added. The mixed solutions were stirred at room temperature for 3 h and then refluxed at 160 °C for 3 h. The products were washed by deionized water and ethanol, then were dried under vacuum at 60 °C for 10 h. The synthesized catalysts were named Pt/Co-NCNT-HPC600, Pt/Co-NCNT-HPC700, Pt/Co-NCNT-HPC800, and Pt/Co-NCNT-HPC900, respectively. The preparation process of Pt/Co-NCNT-HPC catalysts is illustrated in Figure 1.

### 2.4. Characterization

The compositions, structures and morphologies of samples were examined using a Bruker D8 advance X-ray diffractometer (XRD) (Bruker AXS GmbH Inc., Karlsruhe, Germany), a Hitachi SU8010 field emission scanning electron microscope (FESEM) (HITACHI Inc., Tokyo, Japan) equipped with X-ray energy dispersive spectroscopy (EDS) (HITACHI Inc., Tokyo, Japan), and a FEI F30 high-resolution transmission electron microscopy (HRTEM) (FEI Inc., Hillsboro, OR, USA). The chemical states and elemental compositions of the catalysts were measured by a ESCALAB250X X-ray photoelectron spectroscopy (XPS) (Thermo Fisher Scientific Inc., Waltham, MA, USA). The Raman spectra were performed on a Horiba LabRAM HR Evolution Raman microscope (HORIBA Inc., Montpellier, France). The Brunauer-Emmett-Teller (BET) surface area was studied by using a 3H-2000PM2 instrument (Bei Shi De Co. Ltd., Baoji, China).

### 2.5. Electrochemical Measurements

The CHI 760E electrochemical workstation was utilized to analyze the electrochemical performance of samples with three electrodes. A platinum wire and an Ag/AgCl were used as the counter electrode and the reference electrode, respectively. The working electrode is a glassy carbon of 4 mm diameter. Catalyst ink was obtained by mixing 2.5 mg catalyst with 0.05 mL Nafion binder (5 wt%) and 0.45 mL isopropanol aqueous solution (1:1, *v*:*v*), then the catalyst ink was obtained by ultrasonic dispersion for 30 min. Finally, 5 μL of catalyst ink was loaded to the glassy carbon electrode.

The electrochemical activities and stabilities were evaluated by cyclic voltammetry (CV) and chronoamperometry (i–t) experiments in N_2_-saturated 0.5 mol L^−1^ H_2_SO_4_ and 0.5 mol L^−1^ CH_3_OH solution. Additionally, the electrochemically active surface area (ESCA) experiments were carried out in a N_2_-saturated 0.5 mol L^−1^ H_2_SO_4_ solution. Prior to the CV, chronoamperometry, and ECSA tests, the cell was aerated with highly purified N_2_ for 10 min. The CO stripping experiments were filled with high-purity CO for 10 min, followed by N_2_ for 15 min in a 0.5 mol L^−1^ H_2_SO_4_ solution. The whole of electrochemical tests were conducted at ambient temperature with N_2_ protection. The solvents of the above solutions were deionized water.

## 3. Results and Discussions

The surface morphologies of as-synthesized Co-NCNT-HPC support materials were investigated by SEM. In Figure 1, the SEM images at low magnification of 3 μm scale for as-synthesized carbon materials prepared by pyrolysis at varying temperatures are shown. As shown, when pyrolysis temperatures at 600 °C and 700 °C, the surfaces of samples have irregular wrinkles and depressions, which may be due to incomplete pyrolysis of ZIF at the corresponding temperature [31]. When the temperature is 800 °C, the shape of the sample changed significantly, which means the collapse of the ZIF framework. However, as the temperature increases to 900 °C, the shape of the sample may be damaged due to excessive temperature. Thus, it is visibly noticeable that the SEM images of this magnification for a series of different carbonization temperature conditions clearly show that the carbon support has the optimal morphology at a temperature of 800 °C.

The specific morphologies of generated CNTs are observed by magnifying the above samples to higher magnification at 500 nm scale. According to Figure 2, it is observed that the number of CNTs on the surface increased as the pyrolysis temperature increased. Few CNTs are found on the surface of the samples pyrolyzed at temperatures of 600 °C and 700 °C because CNTs are not enough to grow completely at low temperatures. When the temperature is increased to 900 °C, some agglomeration of CNTs could be seen on the surface of the samples due to the excessive temperature. This is due to the aggregation of metal nanoparticles (NPs) caused by high temperature, which leads to evaporation and catalysis when Zn and Co aggregate during pyrolysis, hence the aggregated growth of CNTs. When the temperature is a moderate 800 °C, it is enough to make the CNTs grow completely and in great quantity. The above SEM images at different magnifications all illustrate that 800 °C is favorable for the generation of CNTs.

The SEM images of Pt/Co-PC800 and Pt/Co-NCNT-HPC800 (Figure 3a,b) show that CNTs and hollow polyhedral morphology could not be formed by pyrolysis of pure ZIF-67 but can be obtained by pyrolysis of ZIF-8@ZIF-67. More detailed results are observed from the TEM images of the samples pyrolyzed at 800 °C (Figure 3c,d). It can be seen that compared with Pt/Co-PC800, the Pt/Co-NCNT-HPC800 has a hollow polyhedral morphology with a rough surface, and the surface is covered with anchoring CNTs, which may be because the gas generated during ZIF-8@ZIF-67 pyrolysis drives the metal ions to migrate outward, thus forming a hollow structure [32]. The produced Co NPs accelerated the growth of CNTs by evaporating Zn [33], although the boiling point of Zn is nearly 910 °C, there is still evaporation above 800 °C [34]. Co plays a catalytic role in the growth of CNTs [31], and the evaporation of Zn accelerates this process [33], both of which constitute a synergistic effect of ZIF-8@ZIF-67. The structures of as-prepared CNTs showed large surface area, and hollow carbon structures by high temperature pyrolysis promote well mass and charge transport [35], both conducive to the improvement of methanol electrooxidation performance of Pt-based catalysts.

Specific surface area and pore structure of synthesized carbon supports were investigated by nitrogen adsorption-desorption isotherms. In Figure 4a, the adsorption peaks of Co-PC800 and Co-NCNT-HPC800 at lower relative pressure indicate the existence of micropores in carbon supports [36]. The presence of hysteresis loops between relative pressure of 0.05 and 0.9 suggests the presence of mesopores [37]. The specific surface area and pore volume of Co-PC800 and Co-NCNT-HPC800 are 308.89 m^2^ g^−1^ and 0.30 cm^3^ g^−1^; 323.98 m^2^ g^−1^ and 0.51 cm^3^ g^−1^, respectively. The detailed porosity parameters of the two carbon supports are given in Figure 4b and Table 1. The pore size distribution of Co-PC800 is concentrated, while the pore size distribution of Co-NCNT-HPC800 becomes wider. The average pore size of Co-NCNT-HPC800 and Co-PC800 are 6.26 and 3.92 nm, respectively, resulting in no large macropores in both. The above results can be presumed due to the addition of ZIF-8, which allows the Co-NCNT-HPC800 to have larger specific surface area because more evaporation of Zn atoms can occur, and more mesopores can be generated when the material is pyrolyzed under the same conditions, which are beneficial to the enhancement of electrochemical performance.

The crystalline phase of a series of Pt/Co-NCNT-HPC catalysts was studied with powder XRD measurement in Figure 5. Two major peaks are identified at 39.7° and 46.3°, which are assigned to planes (111), (200) of the face-centered cubic (fcc) of Pt (JCPDS No. 65-2868), which indicates that PtCl_6_^2−^ precursor has been reduced to metal Pt^0^ by EG [38]. Though the other three peaks located at 44.3°, 51.5°, and 76.1° are characteristic of (111), (200) and (220) planes of fcc structure Co (JCPDS No. 15-806) [39]. Except for the weak peak at 26.5°, due to the amorphous carbon of the Co-NCNT-HPC support (planes C (002)) [40], two low intensity peaks located at about 43° and 53°are assigned to planes C (101) and C (004) of the CNT [41]. The first two peaks of Co (111) and Co (200) mask the C (101) and C (004) peaks of CNT with similar peak position and low intensity. The appearance of Co diffraction peaks suggests the reduction of the cobalt ions in the ZIF-67 to Co NPs by pyrolysis. Through a series of XRD patterns, it can be observed that the diffraction peaks of the samples sharp gradually with the increase of pyrolysis temperature, indicating the crystallinity increases by the increase of pyrolysis temperature. However, Co NPs will agglomerate if the temperature is too high [42], which may lead to inhomogeneous physical and chemical properties of the support, and in turn affects the distribution of Pt on the support. In a series of pyrolysis temperatures, 800 °C is the optimized pyrolysis temperature.

The structural changes of the as-prepared Pt/Co-NCNT-HPC catalysts were further proved by the Raman spectroscopic data (Figure 6). The Pt-based catalysts have the similar peaks. They all have D band and G bands around 1340 cm^−1^ and 1597 cm^−1^, respectively [43]. The D bands and G bands reflect graphitic carbon and vibration of the sp^2^ bonded carbon atoms, respectively. Therefore, the degree of defectiveness of Pt/Co-NCNT-HPC catalysts can be quantified by using the area ratio (I_D_/I_G_) of the D to G bands [44]. The I_D_/I_G_ ratio of the Pt/Co-NCNT-HPC600, Pt/Co-NCNT-HPC700, Pt/Co-NCNT-HPC800 and Pt/Co-NCNT-HPC900 are 2.766, 2.995, 2.798 and 2.309, respectively. The I_D_/I_G_ ratio is increasing reasonably because the pyrolysis leads to the breakage of C=C [44]. However, the I_D_/I_G_ ratio decreases when the temperature rises to more than 800 °C, probably due to the higher temperature making the material graphitizing. The Pt-based catalyst supported on carbon material has the highest I_D_/I_G_ value calculated from the area at a pyrolysis temperature of 700 °C, while the highest I_D_/I_G_ value calculated from the intensity ratio is obtained at 800 °C, indicating Pt/NCNT-HPC700 and Pt/NCNT-HPC800 may have the more defective sites, which is conducive to the superior electrocatalytic performance of catalysts.

The chemical states and elemental compositions of as-prepared Pt/Co-NCNT-HPC catalysts were characterized by XPS. Figure 7 shows the XPS spectra of samples. Seen from Figure 7a, the XPS measurement spectra are obtained for different composite catalysts to detect the same peaks, including Pt 4f, C 1s, O 1s, N 1s, and Co 2p. In Figure 7b, the C 1s peaks are segmented into C–C, C–N, C–O–H, C–O–C, C=O, and O=C–O in all samples, which are located at 284.8, 286.1, 287.2, 288.1, 289.3, and 291.2 eV, respectively.

As shown in Figure 7c, the Pt 4f band for catalysts included three pairs of doublets. The first bimodal: the Pt 4f_7/2_ peak at 71.5 eV and the Pt 4f_5/2_ peak at 74.8 eV are assigned to Pt^0^. While the second bimodal: the Pt 4f_7/2_ peak at 72.3 eV and Pt 4f_5/2_ peak at 75.7 eV are assigned to Pt^2+^. Then the third bimodal: the Pt 4f_7/2_ peak at 73.6 eV and Pt 4f_5/2_ peak at 76.9 eV are assigned to Pt^4+^. The Pt contents in the samples were plotted as a histogram displayed in Figure 7d. The content of Pt^0^ in Pt/Co-NCNT-HPC800 is 66.59%, higher than that in other composite catalysts of Pt/Co-NCNT-HPC600 (62.04% Pt^0^), Pt/Co-NCNT-HPC700 (63.55% Pt^0^), and Pt/Co-NCNT-HPC900 (65.34% Pt^0^). The catalyst Pt/Co-NCNT-HPC800 shows the highest Pt^0^ proportion. The above results indicate that the as-prepared porous carbon support (Co-NCNT-HPC800) helps to prevent the re-oxidation of Pt^0^, which results in the enhanced methanol oxidation performance. 

Figure 7e,f show the spectrum of N 1s and the content of N elemental in all samples, indicating the effect of different pyrolysis temperatures on nitrogen doping with a nitrogen source from the organic ligand 2-methylimidazole of ZIF-8@ZIF-67. The N 1s spectrum shows the peak centered at 398.9 eV, 399.0 eV and 401.8 eV, which are assigned to pyridine-N, Co-Nx and pyrrole-N, respectively. Among them, The Co-Nx bond serves as a good catalytic site and has a significant impact on electrochemical catalytic activity [45]. The Co-Nx proportion in Pt/Co-NCNT-HPC800 is determined to be 19.31%, which is higher than that in other catalysts of Pt/Co-NCNT-HPC600 (17.36%), Pt/Co-NCNT-HPC700 (12.46%), and Pt/Co-NCNT-HPC900 (14.78%). The high content of Co-Nx bond for Pt/Co-NCNT-HPC800 means that more active sites are available to enhance the activity of the catalyst. Moreover, the electronic interaction between the nitrogen atoms and Pt NPs allows the nitrogen doping to significantly improve the electrochemical performance for methanol oxidation [46]. Pyridine-N denotes the N atoms in the plane edge of graphite, and pyridine-N represents the N atoms in graphite bonded with three adjacent carbon atoms, respectively [47]. As can be seen from the statistics in Figure 7f, the pyrrole-N proportion in Pt/Co-NCNT-HPC600, Pt/Co-NCNT-HPC700, Pt/Co-NCNT-HPC800 and Pt/Co-NCNT-HPC900 are determined to be 33.66%, 31.69%, 46.79% and 36.36%, respectively. The composite catalyst Pt/Co-NCNT-HPC800 contained the highest pyrrole nitrogen content in a series of samples. Comparatively, pyridine-N has a better charge mobility and superior donor-acceptor charge transferability. Additionally, pyrrole-N can enhance electrocatalytic activity by better donor-acceptor properties and as a good anchor site for Pt [46,48]. So that, increasing the content of pyrrole-N can improve the methanol oxidation performance of the catalyst. Therefore, electrocatalytic performance of catalysts could be enhanced by changing pyrolysis temperatures to increase the pyrrole-N contents of carbon supports.

Figure 8 shows HRTEM images of Pt/Co-PC800 and Pt/Co-NCNT-HPC800 composite catalysts. The distribution and lattice of Pt NPs can be obtained by size distribution histogram and HRTEM image, which are statistics at the 20 nm and 5 nm scale. When comparing this with Figure 8a,c, Pt NPs in Pt/Co-PC800 is agglomerated, while Pt NPs in Pt/Co-NCNT-HPC800 are more uniformly dispersed. The average diameters of Pt NPs in Pt/Co-PC800 and Pt/Co-NCNT-HPC800 are 4.02 nm and 2.91 nm, respectively. The significantly reduced particle size is conducive to exposure of more active sites to improve the electrocatalytic performance [49]. The Pt nanoparticle size of the Pt/Co-NCNT-HPC800 catalyst is significantly smaller than that of Pt/Co-PC800. The well dispersion of Pt atoms and reduction of particle size are due to the large amount of pyrrole-N in the Pt/Co-NCNT-HPC800 sample, which has an anchoring effect as analyzed in XPS, thus avoiding the aggregation and growth of Pt atoms. In addition, HRTEM images from Figure 8b,d show that the lattice spacing of Pt/Co-PC800 and Pt/Co-NCNT-HPC800 is 0.226 nm and 0.204 nm, corresponding to the fcc Pt (111) and Co (111) crystal planes, respectively.

To investigate the electrochemical performance of Pt/Co-NCNT-HPC800, adopt DMFC as the model system, comparing with Pt/C (20%), Pt/Co-PC800, Pt/Co-NCNT-HPC600, Pt/Co-NCNT-HPC700 and Pt/Co-NCNT-HPC900 (Figure 9). The CV curves in 0.5 mol L^−1^ H_2_SO_4_ of catalysts are shown in Figure 9a, indicating the expected behavior of Pt electrodes in the H_2_SO_4_ solution. The area of the electrode in which electrons are exchanged at the electrode is called the active area, or electrochemical surface area (ECSA), which provides an understanding of the number of active sites. The results of ECSA are derived from the equation ECSA = Q_H_/(0.21·L_Pt_), with 0.21 (mC·cm^−2^) as the charge required to oxidize a monolayer of hydrogen, L_Pt_ (mg·cm^−2^) representing the Pt load on the working electrode and Q_H_ (mC·cm^−2^) representing the charge-exchanged during electro-desorption of hydrogen. The ECSA results are reflective of the intrinsic activity of the composite catalyst [50]. The calculated ECSA of Pt/Co-PC800, Pt/Co-NCNT-HPC600, Pt/Co-NCNT-HPC700, Pt/Co-NCNT-HPC800, and Pt/Co-NCNT-HPC900 are 38.52 m^2^ g^−1^, 41.0 m^2^ g^−1^, 43.0 m^2^ g^−1^, 49.7 m^2^ g^−1^ and 45.8 m^2^ g^−1^, respectively, which are larger than 20% Pt/C (27.0 m^2^ g^−1^) and Pt/Co-PC800 (38.5 m^2^ g^−1^). Apparently, from the H_ad_ peak areas of all catalysts, the peak areas of catalysts supported on Co-NCNT-HPC are larger than those of other catalysts, indicating that the catalysts by nitrogen doping and hollow porous possess a larger ECSA. The maximum ECSA value of Pt/Co-NCNT-HPC800 indicates the electrochemically active surface area of a series of Pt/Co-NCNT-HPC catalysts can be increased by appropriate pyrolysis temperature. The highest ECSA value of Pt/Co-NCNT-HPC800 composite catalyst can be responsible for the small particle size and well dispersion of Pt atoms on the support, as well as a large number of CNTs with good internal channels, which can facilitate the transport of electrons and become more conducive to the exposure of more active sites.

Figure 9b shows the CV curves of a series of Pt/Co-NCNT-HPC catalysts in N_2_-saturated 0.5 mol L^−1^ H_2_SO_4_ and 0.5 mol L^−1^ CH_3_OH solution to test methanol oxidation activity. The current densities of different samples are normalized by Pt loading (mass activity). The peak level reaches a maximum current density at about 0.7 V when CV performs a positive sweep and at about 0.45 V when it performs a negative sweep, which is related to incomplete methanol oxidation and the intermediate products (CO_ad_) of methanol oxidation, respectively [51,52]. The mass activities of the samples of Pt/Co-PC800, Pt/Co-NCNT-HPC600, Pt/Co-NCNT-HPC700, Pt/Co-NCNT-HPC800 and Pt/Co-NCNT-HPC900 are 295.4, 162.9, 249.8, 416.2 and 334.8 mA mg^−1^_Pt_, respectively. The results show the mass activity of the composite catalyst supported on Co-NCNT material is higher than that of 20% Pt/C (153.6 mA mg^−1^_Pt_), especially the Pt/Co-NCNT-HPC800 catalyst having the highest mass activity. The mass activity of Pt/Co-NCNT-HPC800 catalyst for DMFC is ~2.7 times and ~1.4 times higher than that of commercial Pt/C and Pt/Co-PC800 catalyst, respectively. This excellent performance might be due to the smaller Pt particle size, which is consistent with the HRTEM result in Figure 8 and its larger ECSA.

Figure 9c shows a typical set of CO-stripping voltammograms cyclic (dashed curve) and without (solid curve) CO_ad_ at Pt-based catalyst. During the electrocatalytic oxidation of methanol part of the catalyst, Pt will undergo an indirect dehydration pathway to form easily adsorbed reaction intermediates CO_ad_, which can occupy the active site of Pt to make it poisoned, thus preventing the further oxidation of methanol and leading to a decrease in catalytic performance, so to evaluate the CO poisoning resistance of the prepared Pt-based composite catalysts and provide valuable information about the nature of the samples. This peak disappears in the second cycle (solid curve) which means the CO_ad_ is oxidized almost completely in the first cycle (dashed curve) and the synthesized catalyst has very little affinity with CO. A lower onset potential implies the intermediate CO_ad_ is more easily oxidized, thus releasing the catalyst active site [51]. The CO_ad_ oxidation onset potentials of Pt/C (20%), Pt/Co-PC800, Pt/Co-NCNT-HPC600, Pt/Co-NCNT-HPC700, Pt/Co-NCNT-HPC800 and Pt/Co-NCNT-HPC900 are 0.64, 0.60, 0.58, 0.57, 0.51 and 0.51 V, respectively. Compared with 20% Pt/C and Pt/Co-PC800, Pt/Co-NCNT-HPC, a catalyst with nitrogen-doped carbon nanotube hollow porous carbon support has the lower CO_ad_ oxidation onset potential. The results show that the adsorption strength and amount of CO_ad_ can be reduced by nitrogen-doped carbon nanotube porous carbon support. The Pt/Co-NCNT-HPC800 and Pt/Co-NCNT-HPC900 have the lowest CO_ad_ oxidation onset potentials. Compared with Pt/C (20%), the peak potential of CO_ad_ oxidation on Pt/Co-NCNT-HPC800 and Pt/Co-NCNT-HPC900 shifted negatively about 130 mV, indicating the pyrolysis temperature need above 800 °C. The corresponding catalysts Pt/Co-NCNT-HPC800 and Pt/Co-NCNT-HPC900 have the similar CO_ad_ stripping potential compared with Pt/C, indicating that the pyrolysis temperature above 800 °C has little effect on CO poisoning resistance. Therefore, increasing the pyrolysis treatment temperature to 800 °C makes the CO_ad_ adsorption strength weaker and the CO_ad_ oxidation kinetic activity stronger, which is beneficial to improving the electrochemical performance of catalysts for methanol oxidation.

Improving the durability of catalysts is crucial for their practical application in DMFCs. Chronoamperometry i–t is an effective technique to evaluate the durability of catalysts for DMFC, in which the potential is fixed at 0.7 V vs. Ag/AgCl, which is the typical working potential in DMFCs. Relative to Figure 9d, all catalysts showed the same trend of mass activity change for MOR, which all decreased rapidly in the initial stage and kept at a relatively stable level after a sharp decrease. The current reduction reflects the poisoning caused by the adsorption of CO_ad_ on the Pt surface during MOR. The electrocatalytic activities of Pt/C (20%), Pt/Co-PC800, Pt/Co-NCNT-HPC600, Pt/Co-NCNT-HPC700, Pt/Co-NCNT-HPC800 and Pt/Co-NCNT-HPC900 for MOR at 500 s are 4.86, 13.47, 14.56, 17.45, 69.34 and 32.76 mA^−1^_Pt_, respectively. After 3600 s, the Pt/Co-NCNT-HPC800 catalyst still maintained the highest mass activity at 27.18 mA mg^−1^_Pt_, which is 10 times as high as that of the Pt/C (20%). Furthermore, in a series of Pt/Co-NCNT-HPC catalysts, the mass activity of Pt/Co-NCNT-HPC800 catalyst is much higher than that of others over the entire scan time range. The result shows the Pt/Co-NCNT-HPC800 catalyst has much higher durability than that of other catalysts. This proves that the Co-NCNT-HPC support obtained by high-temperature pyrolysis at a suitable temperature can effectively improve the stabilities of Pt-based composite catalysts. The Pt/Co-NCNT-HPC800 has the best long-term stability for methanol electrooxidation owing to its relatively highest specific surface area and high ECSA value, which assists with removing efficiently the intermediate poisonous species. Detailed electrochemical data of Pt-based composite catalysts are shown in Table 2.

## 4. Conclusions

In summary, a series of Pt-based catalysts supported on ZIF-8@ZIF-67-derived Co-NCNT-HPC were successfully synthesized. The presence of ZIF-8@ZIF-67 is found to be critical as they formed Co-NCNT-HPC materials. Among them, Zn in ZIF-8 forms pores during high temperature pyrolysis, as well as Co in ZIF-67 and Zn in ZIF-8 constituting a synergistic effect, which together form hollow structures and derived CNTs favorable for specific surface area increase and mass and charge transport. Moreover, the pyrolysis of the organic ligands 2-methylimidazole of ZIF-8 and ZIF-67 doped the carbon material support with nitrogen, and the appropriate pyrolysis temperature (800 °C) gave support to the highest pyrrole-N concentration, which acted as an anchor for Pt. In summation, the role of ZIF-8@ZIF-67 is fully exploited in the process of improving the performance of methanol oxidation. The Pt/Co-NCNT-HPC800 catalyst has the larger specific surface area, the smaller particle size, and good dispersion. Hence, the Pt/Co-NCNT-HPC800 catalyst exhibits excellent mass activity, CO poisoning resistance and electrocatalytic stability toward methanol oxidation. This work offers promising insights into the design of DMFC catalyst for electrocatalysis application.

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
