# Peer review of "ZIF-8@ZIF-67-Derived Co Embedded into Nitrogen-Doped Carbon Nanotube Hollow Porous Carbon Supported Pt as an Efficient Electrocatalyst for Methanol Oxidation"

_nanomaterials, 2021, doi:10.3390/nano11102491_

Round 1
Reviewer 1 Report
The manuscript entitled 'ZIF-8@ZIF-67-Derived Nitrogen-Doped Carbon Nanotube Hollow Porous Carbon Supported Pt as an Efficient Electrocatalyst for Methanol Oxidation' (Ms. ID Nanomaterials-1377168) reports the use of a mixed MOF (ZIF-8 and ZIF-67) for the formation of a porous carbon structure doped with N to use as support of Pt nanoparticles. The resulting catalyst has been evaluated for the electrocatalytic oxidation of Methanol. Temperature influence during the formation of the carbon material has been studied.
1) Why the combination of ZIF-8 and ZIF-67 conducts to a better catalyst? What about the catalysts from the pure MOFs? This aspect is not explained nor studied.
2) Line 181. Zn is evaporated. Zn evaporates at nearly 910ºC and the temperature of the optimum catalyst is 800ºC. How to ensure that there is no Zn in the solid. Specific analysis are required.
3) What is the role of Co particles in the final catalyst? How does Co influence in the electrocatalytic oxidation? The material without Pt should be assessed.
4) I miss a termogravimetric analysis since it is relevant for the optimization of the temperature during the formation of the carbonaceous material.
5) English should be immproved.
Reviewer 2 Report
Dear Authors,
The topic of the article: ZIF-8@ZIF-67-Derived Nitrogen-Doped Carbon Nanotube Hollow Porous Carbon Supported Pt as an Efficient Electrocatalyst for Methanol Oxidation, is interesting. However, I have many reservations and doubts regarding the conduct of electrochemical tests, which, in my opinion, should be repeated so that the results of your tests can be published.
I present my comments below:
XRD patterns of Pt / NCNT-HPC catalysts
it would be reasonable for comparison to present data for the carrier itself, because the signals observed in Figure 5 are consistent with the signals coming from carbon nanotubes, i.e. about 26 °, 43 °, 53 °.
Raman Spectroscopy
I do not think it is reasonable to determine the Id / Ig ratio from the signal height, because on the recorded spectra the signals are significantly broadened. In this case the ratios should be taken from the integrated band areas, which are best derived by curve fitting spectra that have no background contribution.
Ad electrochemical measurments
- In what solvent the tests were conducted?
- What and in what concentration was the electrolyte in the Ag / AgCl electrode?
- in Figure 5 in panel c the axis x is not described, the description of the drawing itself is incorrect "(c) Amperometric i-t curves in the amount of 0.5 mole L-1 H2SO4 and 0.5 mol L-1 CH3OH solution; "this figure shows CV data.
- What is the influence of Nafion on the behavior of the Catalytic system?
- The descriptions of Fig. 9b and Fig. 9c imply that the measurements were conducted in the same environment (containing MeOH and H2SO4), with Fig. 9c also containing data recorded in the presence of CO. Why, therefore the data presented in Fig. 9c without the presence of CO are so vastly different from those presented in Fig. 9b?
- What processes are the two oxidation signals observed in Fig. 9b attribted to? Why are oxidation signals present in both the anodic and cathodic half-cycles? What was the direction of potential changes in the presented CVs?
- Was the working electrode one with a certified and constant Surface area? If not, how were the current density values calculated in the presented figures?
- Why were the current densities normalised to Pt loading instead of taking into account also the Surface area of the electrodes?
- The determination of the electorchemical Surface area should be explained in detail.a
Round 2
Reviewer 1 Report
Afther addressing the reviewers' comments, I feel that the Ms. can be accepted for publication.
Reviewer 2 Report
The Aurors addressed all my comments and reservations. In my opinion, the article may be published in MDPI Nanomaterials.